# Identification of Oocyst-Driven *Toxoplasma gondii* Infections in Humans and Animals through Stage-Specific Serology—Current Status and Future Perspectives

**DOI:** 10.3390/microorganisms9112346

**Published:** 2021-11-13

**Authors:** Gema Álvarez García, Rebecca Davidson, Pikka Jokelainen, Siv Klevar, Furio Spano, Frank Seeber

**Affiliations:** 1Saluvet Group, Animal Health Department, Complutense University of Madrid, Ciudad Universitaria s/n, 28040 Madrid, Spain; gemaga@ucm.es; 2Department of Animal Health and Food Safety, Norwegian Veterinary Institute, 9016 Tromsø, Norway; rebecca.davidson@vetinst.no; 3Department of Bacteria, Parasites and Fungi, Infectious Disease Preparedness, Statens Serum Institut, 2300 Copenhagen, Denmark; PIJO@ssi.dk; 4Department of Analysis and Diagnostics, Norwegian Veterinary Institute, 1433 Ås, Norway; siv.klevar@vetinst.no; 5Unit of Foodborne and Neglected Parasitic Diseases, Department of Infectious Diseases, Istituto Superiore di Sanità, 00161 Rome, Italy; furio.spano@iss.it; 6FG 16: Mycotic and Parasitic Agents and Mycobacteria, Robert Koch-Institut, 13353 Berlin, Germany

**Keywords:** antigen prediction, oocyst-specific antigens, stage-specific serology, toxoplasmosis, source attribution, surfaceome

## Abstract

The apicomplexan zoonotic parasite *Toxoplasma gondii* has three infective stages: sporozoites in sporulated oocysts, which are shed in unsporulated form into the environment by infected felids; tissue cysts containing bradyzoites, and fast replicating tachyzoites that are responsible for acute toxoplasmosis. The contribution of oocysts to infections in both humans and animals is understudied despite being highly relevant. Only a few diagnostic antigens have been described to be capable of discriminating which parasite stage has caused an infection. Here we provide an extensive overview of the antigens and serological assays used to detect oocyst-driven infections in humans and animals according to the literature. In addition, we critically discuss the possibility to exploit the increasing knowledge of the *T. gondii* genome and the various ‘omics datasets available, by applying predictive algorithms, for the identification of new oocyst-specific proteins for diagnostic purposes. Finally, we propose a workflow for how such antigens and assays based on them should be evaluated to ensure reproducible and robust results.

## 1. Source Attribution of *Toxoplasma gondii* Infections Is Challenging but Relevant

### 1.1. A Quick Tour through the Life Cycle of Toxoplasma gondii

The apicomplexan protozoan *Toxoplasma gondii* is a highly successful cosmopolitan intracellular parasite of the Sarcocystidae family, which can cause a range of disease manifestations in humans and animals [1]. The parasite has an indirect predator–prey life cycle and a wide host range, with felines as the definitive host [2]. It is able to invade any nucleated cell in warm-blooded animals, including humans, which can serve as intermediate hosts. Infection with *T. gondii* has gone from being considered mostly benign, with risk being limited to acute infections during pregnancy and reactivation of chronic infections in immunosuppressed individuals, to a much greater public health concern [3]. When using disability-adjusted life years as a measure of disease burden, congenital toxoplasmosis is ranked alongside hepatitis B and pneumococcal infection and ahead of tetanus in Europe [4]. Ocular toxoplasmosis is among the other manifestations contributing to the disease burden [5], and chronic *T. gondii* infections have been associated with mental disorders, although solid evidence for this is limited and requires further studies [6].

*Toxoplasma gondii* can be transmitted by three different developmental stages (infection routes): tachyzoites (congenital), bradyzoites within tissue cysts (meat-borne pathway), and sporozoites within sporulated oocysts (environmental pathway). All three stages can result in patent infections in the definitive hosts, felids, and in tissue infections in intermediate hosts [7,8]. While infected felids as the definitive hosts are able to shed massive numbers of oocysts via their feces into the environment [9], the parasite is also able to circulate between intermediate hosts through asexual reproduction [10,11], even in the absence of definitive hosts [12]. The longevity of tissue cysts in different host species is not certain but assumed to last in most cases as long as the lifetime of the host [13]. Congenital transmission, from mother to fetus, also occurs in a wide range of hosts, including humans, sheep and rodents (reviewed in [14]). Less frequent but important transmission routes include organ transplantation and blood transfusion [15].

Once ingested, the infective sporozoites or bradyzoites enter the intestinal epithelium to reach the lamina propria from where, after differentiation into tachyzoites, the parasites disseminate throughout the whole body, with tissue cysts showing a marked predilection for skeletal and cardiac muscle as well as neural and ocular tissue [16,17]. After host cell invasion, the tachyzoites proliferate within a parasitophorous vacuole in the host-cell cytoplasm, and ultimately egress and infect neighboring cells.

*Toxoplasma gondii* induces a strong inflammatory response by the host, which plays a critical role in controlling the infection and limiting parasite burden. Protective immunity to *T. gondii* involves both the innate and the adaptive immune response [18]. It is dominated by antibody production against parasitic antigens, and is primarily dependent on T helper 1 cell-mediated immunity which is characterized by high levels of interleukin-12 and interferon-γ. Besides controlling the acute infection, these cytokines also induce tissue cyst formation of the slowly replicating bradyzoites and thus enable sustained latent infection [19]. Recrudescent infection can occur if the immune status of the host is compromised, resulting in conversion of bradyzoites back to the tachyzoite stage.

### 1.2. What Is the Relative Importance of Meat-Borne vs. Oocyst-Driven Transmission of T. gondii?

Toxoplasmosis is a significant public health problem worldwide and qualifies as a One Health disease because it significantly affects the health and well-being of humans, domestic animals, wildlife, and ecosystems [20,21]. It is estimated that globally roughly one third of the human population has latent *T. gondii* infections, with considerable regional variations in prevalence [22]. In livestock infections can also be frequent. *T. gondii*-associated abortions in sheep are generally attributed to recent oocyst exposure, and control measures are focused on biosecurity procedures and vaccines where available [23]. In pigs, it has been traditionally accepted that clinical infection is unapparent and transplacental transmission is infrequent [24]. However, *T. gondii* has been identified as a cause of reproductive disorders in sows and an increasing number of studies have reported outbreaks of clinical toxoplasmosis in fattening pigs [24,25].

Experimental data have shown that ingestion of tissue cysts is the most efficient transmission route for cats whilst for intermediate hosts (mice, rats, small ruminants, and pigs) this seems to be infection by oocysts [26,27,28]. The substantial infection rates seen in herbivores would support the high infectivity of the oocyst stage for other animals as well [29,30], consistent with the hypothesis that “*T. gondii* is biologically adapted to transmission by carnivorism in cats and by fecal–oral route in herbivores” [28]. The situation is less clear for the omnivorous humans. Although it is widely accepted that the majority of human infections in industrialized countries occur via consumption of meat from infected animals, in a multicenter study in Europe 30–60% of infections could be attributed to meat as infection source, and 6–17% to contact with soil (oocysts) [31]. A WHO expert opinion paper estimated that 45–61% of *T. gondii* infections could be attributed to meat-borne transmission whilst environmental transmission via oocysts was also highlighted as an important route of infection [32]. Data available on outbreaks of acute toxoplasmosis revealed that 47.1% of outbreaks were related to tissue cysts and 44.1% were related to oocysts [33]. It is worth mentioning that following increased stormwater runoff events, which are expected to become more common due to climate change, the spread of oocysts far away from cat defecation spots can occur [34]. Together with fairly high *T. gondii* seroprevalences reported in strictly vegetarian human sub-populations [35,36,37], and the low seroprevalence on islands free of cats [38], the documented outbreaks clearly indicate that oocysts are an important source for infection also for humans. However, no solid data exist regarding the minimum infection dose of oocysts for humans nor the degree of the environmental contamination and thus exposure risk for humans and animals [39].

Food and waterborne routes of *T. gondii* infection have received increased interest in recent years (reviewed in [15,40,41]). Foodborne outbreaks have been traced to the consumption of meat of infected animals, fresh produce and milk products. The consumption of oocyst-contaminated food products such as shellfish (whereby filter feeders act as mechanical vectors) as well as water have also been implicated in outbreaks or as risk factors for infection [42,43]. There are few documented *T. gondii* outbreaks in humans, and the acute infection can present with vague, unspecific clinical signs that can be overlooked [5]. Waterborne outbreaks have been easier to identify given the large number of people affected [33,40,44,45].

## 2. The Challenge of Differentiating between Meat-Borne and Oocyst-Driven *T. gondii* Infections

The relative importance of different *T. gondii* transmission pathways is difficult to assess and, from a One Health perspective, represents a major gap that needs to be addressed to implement future intervention strategies [15]. Conventional serological tests to diagnose postnatally acquired *T. gondii* infection do not discriminate between meat-borne and environmental pathways. The meat-borne pathway has been traditionally assumed to predominate in industrialized countries, whereas more cases of environmental infections have been described in developing countries. However, over the last few years an increase of toxoplasmosis cases has coincided with a higher consumption of fresh foods such as fruits and vegetables, in particular as ready-to-eat products [15]. Moreover, knowing the ways omnivorous animals that are raised for human consumption become infected with the zoonotic parasite would enable targeted interventions at farm-level. There is a need, therefore, to distinguish between tissue cyst- and oocyst-driven infections in both humans and animals. Such an approach would help to: (i) evaluate the relevance of the different transmission routes, in particular to assess the contribution of oocyst-driven infections; (ii) assess whether the clinical outcomes are different depending on the route of transmission; and (iii) prioritize targeted interventions. Oocyst-driven infections of humans have scarcely been studied and are likely underreported, with documented outbreaks only reported from a few countries (e.g., Brazil) [33]. There is non-conclusive evidence of an association between the parasite stage ingested and the severity of toxoplasmosis in humans [40].

### 2.1. Overview of Serological Tests for the Diagnosis of T. gondii Infections in Humans and Animals

Serology is a crucial tool for the diagnosis of *T. gondii* infection and there is extensive literature addressing the use of a wide battery of serological tests employed in humans and animals [46,47,48,49]. Traditionally, most serological tests have been used for the purpose of differentiating seropositive vs. seronegative individuals (i.e., having antibodies against the parasite or not), but serology can also distinguish acute from chronic infection. Limited attempts have been carried out for using serology to identify infection sources, and there is only a single example that used an extensive One Health approach, in which environmental contamination with oocysts was studied paying attention to humans, animals and drinking water [50].

Differentiating acute vs. chronic infection helps to estimate the time of infection and is a vital part of epidemiological outbreak investigations to discriminate between potential sources and routes of infection. Approaches include classical pairwise IgM/IgG comparisons and IgG avidity testing, which have been well implemented in the human medical field, in particular for pregnant women. A range of commercial assays are available [49]. Serological assays based on combinations of recombinant proteins have also been adapted to differentiate acute from chronic *T. gondii* infections. For example, parasite proteins MAG1, GRA2, GRA6, GRA7 and ROP1 have been suggested to indicate acute *T. gondii* infections (reviewed in [51]) and GRA5 chronic infections [52]. Less progress has been made in diagnosing acute vs. chronic infections in animals. Experimental techniques and assays have been developed (see Appendix A for further details and discussion) but are rarely employed.

In humans, detection of specific IgM antibodies is considered an early marker for acute phase of the infection, but their presence should be interpreted with caution as they can persist for 18 months after the infection (for more information see [1,48,53,54]). Acute infections, when accompanied with appropriate epidemiological data (diet, environmental exposure), may direct suspicion to oocysts as the source of infection.

Limited information is available on relevant antibody dynamics in livestock based on experimental infections with oocysts (see Appendix A; [55,56,57,58,59,60,61]).

### 2.2. Investigating the Route of Infection—What Assays Are Available So Far?

It has been reasonable to propose that proteins expressed exclusively by *T. gondii* oocysts may help to identify the early phase of an oocyst-driven infection [62,63,64]. The interactions between the host and the wall of the oocyst, the sporocysts and the motile sporozoites are short-lasting. In fact, intact oocysts and sporocysts (or the remnants of their walls upon excystation) are expected to undergo quick transit through the host intestines, and the sporozoites are known to differentiate into tachyzoites as early as 12 h p.i. Therefore, assuming a low likelihood of host re-exposure to oocysts within a short time, this parasite stage is expected to induce a low level of antigenic stimulation, as suggested by some studies [62,64,65]. Moreover, the question how extracellular particulate oocyst constituents, such as membrane-bound oocyst wall proteins, located in the intestinal lumen, reach antigen-presenting cells in the lamina propria has been raised and needs to be addressed [66]. Thus, the likely absence of a durable immune response against oocyst antigens in the absence of booster stimulations represents a major challenge. This might hamper their diagnostic use as markers of oocyst exposure. What follows is a discussion of the few antigens that have, so far, been reported as being used for this task in humans and animals.

### 2.3. What Antigens Are Already Described for Identification of Oocyst-Driven Infections?

#### 2.3.1. Oocyst Wall Proteins

The oocyst wall of *T. gondii* is an elaborate and highly resistant two-layered shell, granting long term environmental survival to the sporozoites. Following pioneering ultrastructural studies [67,68], major insights into the molecular composition of the oocyst wall were obtained in the last decade. Besides the fungal cell wall-related polysaccharide beta-1,3-glucan [69] and a coat of acid-fast lipids [70], several proteins involved in the structural organization of the inner and outer wall have been described [71,72]. The first such proteins of *T. gondii* were identified by searching the ToxoDB database for homologs of the *Cryptosporidium parvum* oocyst wall protein COWP1 [73]. Of the newly identified cysteine-rich proteins, dubbed TgOWP1-7 [71], TgOWP1-3 have been localized to the oocyst wall, and TgOWP3 has specifically been assigned to the outer layer. TgOWP1 was used in serological studies but with limited success due to its low antigenicity [64]. While it was reported as a marker of oocyst-driven infection in chickens, no reactivity against this protein was detected in similarly infected pigs. Whether TgOWP1 is immunogenic in humans still needs to be explored.

Additional cysteine-rich members of the *T. gondii* OWP family, designated TgOWP8-12, have been described and oocyst wall localization was demonstrated for TgOWP8 [72]. The diagnostic value of this antigen has been tested recently [63]. Results obtained by Western blot and ELISA tests showed that TgOWP8 seems to be antigenic in humans, pigs and chickens. Similar prevalence rates were obtained in pigs and chickens when TgOWP8 ELISA results were compared with those previously reported using the sporozoite-specific protein CCp5A (see below; [64]). However, prevalence rates should be carefully interpreted since the diagnostic performance of these assays is unknown (Table 1 and Table 2).

#### 2.3.2. Sporozoite Proteins

Over the last decade, three distinct proteins specifically expressed by *T. gondii* sporozoites, namely ERP, SporoSAG and CCp5A, have been reported to be able to detect oocyst-driven infections by serological tests (Table 2 and Table 3).

The ERP protein, which was employed in several seroepidemiological studies [50,62,74,75], is part of a group of four molecules annotated as ‘late embryogenesis abundant domain-containing proteins’ (LEAs). They are potentially involved in conferring stress resistance to the oocyst [77,78], and the encoding genes are significantly upregulated in this parasite stage. The 11 kDa ERP was identified by comparative 2D gel electrophoresis and mass spectrometry of tachyzoite, bradyzoite and sporozoite proteins [62] and proved to be abundantly expressed in the latter stage. ERP was recognized by sera from pigs that were experimentally infected with oocysts and seemed to differentiate oocyst-driven infections vs. tissue cyst-driven infections. It also elicited a specific antibody response in *T. gondii*-infected mice and humans. In the latter, it was reported to identify *T. gondii* seropositive subjects within 6–8 months after oocyst ingestion [62]. The authors suggested ERP to be an early infection marker (despite its recognition by chronically infected humans) due to a unique exposure of the immune system to sporozoites vs. a permanent exposure to tachyzoite or bradyzoite stages. In further studies higher detection rates of ERP by human sera derived from settings with possibly high environmental oocysts contamination were described [50,75], compared to sera from areas where infection by the meat route was considered predominant [74]. Vieira et al. reported the detection of anti-ERP antibodies in chickens and a high prevalence of anti-ERP antibodies in humans, providing evidence for the relevance of environmental oocysts contamination due to underground water contamination [50]. In addition, the dependence on subjects’ age for successful salivary IgA detection against ERP in determining the mode of *T. gondii* transmission in endemic settings was demonstrated [75]. The cut-off estimation of the method significantly influenced the prevalence rates obtained in the different age groups. Moreover, there was poor correlation between IgA salivary and IgG systemic levels, with the exception of the 15–21-year age group. Further work beyond the research group that first described ERP is needed to corroborate the robustness of this protein as an indicator for oocyst-derived infection.

SporoSAG, a member of the large SRS family of proteins tethered to the surface of the different invasive stages via their GPI anchor [79], is abundantly expressed in sporozoites [77,80,81]. It seems to be involved in host cell invasion, so that early exposure to the immune system could support a diagnostic value in early infections. Only two studies have evaluated the antigenicity of SporoSAG, with contradictory results. Crawford et al. reported that this protein was not immunogenic in natural infections since neither anti-SporoSAG IgA, IgM or IgG antibodies were detected in humans after infections with oocysts [76]. Furthermore, sera from experimentally infected mice were unable to detect SporoSAG. In contrast, another study found that in oocyst-infected mice the serum titer of anti-SporoSAG antibodies increased until 40 days p.i. [65]. These discrepant results might be explained by several different experimental conditions (sex, age and strain of mice; inoculation method (oral vs. subcutaneous) and p.i. time points) and expression systems for the production of recombinant SporoSAG (e.g., insect cells vs. *Escherichia coli*). This highlights the need for more standardized regimens for the validation of potential diagnostic antigens.

The third sporozoite-specific protein investigated for its source-attributing potential is CCp5A. It belongs to a family of seven *T. gondii* multi-modular proteins sharing the presence of one or more copies of the so-called LCCL amino acid domain [82] and is localized to unidentified granules scattered throughout the sporozoite (F. Spano; unpublished results). The diagnostic value of CCp5A was studied in different experimentally infected animal species and also in humans [64]. A bacterially expressed 50 kDa recombinant fragment of CCp5A was detected by sera from oocyst-infected animals (pigs, mice and chickens) but not from animals infected with either tachyzoites (pigs) or tissue cysts (mice). Interestingly, human IgM and IgG in sera from a toxoplasmosis outbreak also recognized CCp5A. More recently, Liu et al. also described CCp5A-seropositive humans and chickens using a 17 kDa recombinant fragment of the protein, but the true seroprevalence rates are unknown due to the lack of epidemiological data and serological test validation [63].

Finally, a recent approach confirmed the difficulty in identifying antigens with diagnostic value for the differentiation of infection routes. A panel of more than 2870 genes from *T. gondii*-expressed exon products were probed in a microarray format [65]. Sera from mice infected with either oocysts or tissue cysts were analyzed and the authors recorded a specific IgM response from 10 to 15 days p.i. and a specific IgG response from 10 to 120 days p.i. (the endpoint of the experimental assay). This allowed them to identify a panel of immunogenic proteins with putative diagnostic and/or vaccine value. However, according to the reported IgG response, the same candidates were recognized by both oocyst- and tissue cyst-infected mice. Similarly, IgM responses against GRA6, GRA8 and ROP1 were detected by oocyst-infected mice, but later on these proteins were also recognized by IgGs from all infected animals, regardless of infection source. These results support the need to continue the search for sporozoite- or oocyst-specific antigens for the differentiation of the route of the infection.

## 3. Finding New Antigens for Identifying Oocyst-Driven *T. gondii* Infections by Experimental Approaches and In Silico Antigen Prediction

Early studies used immunoblotting with monoclonal antibodies [83] or infection sera [84] to investigate the existence of *T. gondii* sporozoite-specific surface antigens. More recently, this immunological approach was coupled to protein sequencing to identify the antibodies’ molecular targets in the tachyzoite stage [85,86,87]. Protein arrays, containing polypeptides derived from in vitro-translated exons of *T. gondii* genomic DNA, can be considered a modern version of this method. In this approach the polypeptides are attached to different matrices (membranes, glass slides, beads) and probed with infection sera from animals and humans to identify antigenic proteins. In a series of reports a set of 280 *T. gondii* gene products were identified that, to varying degrees, showed reactivity with human sera from acutely or chronically infected subjects [52,88] or with sera from mice orally infected with either *T. gondii* oocysts or tissue cysts [89].

Peptide arrays follow the same rationale whereby short synthetic peptides derived from known protein sequences are used as antigens. They have been exploited in the past for various pathogens [90,91]. In one approach taken by various groups, arrays were composed of numerous polymorphic peptide sequences of *T. gondii* proteins with the aim to identify those that would allow the differentiation of different clonal lineages [92,93,94,95]. This approach has led to some success, but it mostly focused on proteins already known to be good antigens rather than trying to identify new ones [90].

### 3.1. What Makes a Protein a Good Antigen?

While these studies have provided important insights into a subset of *T. gondii* proteins exposed to the host’s immune system, they have almost all focused on either tachyzoite proteins or did not address the putative bradyzoite or oocyst/sporozoite specificity of the antigens identified.

The following sections describe approaches for their potential in silico identification. It is well accepted, and also backed up by experiments with large-scale protein microarrays, that antigenically relevant proteins from pathogens which are recognized by antibodies (B-cell antigens) should be surface-exposed or extracellular and have: (i) a predicted signal peptide; (ii) 1–10 transmembrane regions; (iii) a prediction for plasma membrane localization; (iv) an isoelectric point between 7 and 9 and (v) a relatively high abundance compared to the average proteome (reviewed in [96]). However, a recent antigen array study, based on the almost complete set of proteins (91%) from *Plasmodium falciparum,* found that also a very large number of proteins predicted or known to be intracellular contributed to a high and diverse level of individual immune reactivities in previously parasite-exposed adults in Tanzania [97]. The authors reasoned that cellular turnover of infected cells might also result in the presentation of intracellular proteins from the parasite to the immune system. This is in contrast to bacterial pathogens where surface exposed proteins predominate in eliciting antibody responses [96,98].

Nevertheless, an important determinant would be a complete knowledge of the surfaceome, i.e., the entirety of all proteins on the cell surface [99]. Experimental approaches for this involve specific labeling of surface proteins (e.g., via biotinylation) and their subsequent enrichment, followed by their identification by mass spectrometry [100]. A single study has reported such data for tachyzoites of *T. gondii* [101] (similar data for bradyzoites and sporozoites are not available). Surprisingly, out of the 247 reported surface-biotinylated proteins that can still be identified in ToxoDB (https://toxodb.org, accessed on 21 July 2021), the major hub for these and other ‘omics’ data provided by the research community [102], none belonged to the 111 annotated SAG-related surface (SRS) proteins. Other features one would expect for proteins transported to the cell surface, such as a predicted signal peptide, GPI anchor or transmembrane regions, are also detectable only in a minority of these molecules (data not shown). This illustrates that it is difficult to judge if they are genuine surface-located proteins and casts doubts on the reliability of this experimental approach.

Proteomic data of oocyst wall proteins were described by Fritz et al. [77]. In this study cell fractionation was the basis of the wall sub-proteome, although sporozoite contamination of the wall-enriched fraction was not reported. A surprisingly high proportion (28%; 62 of 221) of these proteins are in common with the surface-biotinylated tachyzoite proteins from Che et al. [101], indicating absence of stage-specificity.

With regard to the bradyzoite stage, 42 experimentally determined cyst wall proteins have been described [103,104]. Here, a combination of Percoll gradients, subsequent immunoprecipitation with an antibody directed against a known wall protein, CST1, followed by proteomics was used. Of those proteins, 11 (26%) are shared with the above-mentioned surface-biotinylated tachyzoites.

An important advance is the recent experimental allocation of nearly 2000 proteins with hitherto unknown localization within the parasite to different cellular sub-compartments [105], based on a method called hyperLOPIT [106]. The caveat in the context of this review is again that it was performed with tachyzoites. Notably, 1558 proteins from a total of 2647 high-confidence hyperLOPIT-assigned proteins are also detected in oocyst proteomes. Therefore, truly oocyst- or bradyzoite-specific antigens will not be present in this dataset. Two recent algorithms have been described, BUSCA [107] and DeepLoc [108], that allow genome-wide predictions of subcellular localizations of proteins. They might be useful to fill the gaps left by missing hyperLOPIT assignments (but see above [97]).

### 3.2. In Silico Prediction of the Surfaceome—Illustrating the Limits

Currently, more than 60 high-quality genome sequences of different *T. gondii* haplotypes are publicly accessible via ToxoDB [102]. It contains not only the protein sequences deduced from genomic and mRNA sequences but also various proteomes derived from all three *T. gondii* infectious stages, via mass spectrometry. However, the majority of proteomic data comes from several reports that studied various aspects of tachyzoite biology [109]. Numerous quantitative data sets of RNAseq or microarray experiments allow conclusions about stage-specific mRNA abundance of specific genes under different experimental or environmental conditions. Below we describe a number of datasets that were either taken from ToxoDB or compiled by us for the purpose of antigen prediction in an Excel file (Appendix A). We provide a set of 8284 curated proteins of the ME49 strain with corresponding predictions or data compiled from their sequences. Third-generation genome sequencing generates more complete and accurate chromosome data of *T. gondii* strains [110,111] and might add to a more diverse proteomic landscape in the future than reflected by the current data. Intron retention has been widely observed in *T. gondii* oocyst/sporozoite-specific genes, e.g., TgOWPs, SporoSAG, which are transcribed but not fully spliced in tachyzoites or bradyzoites [71,112,113], making accurate protein annotation for these stages troublesome.

A rather simplistic approach for the in silico prediction of the surfaceome could be done by using the scheme from above [96]. Bioinformatic prediction of membrane-bound proteins relies on the detection of transmembrane (TM) regions within the protein sequence. The most widely used programs for this purpose are TMHMM 2.0 [114] and TOPCONS 2 [115], the latter also being able to discriminate hydrophobic signal peptide sequences from true TM helices [116]. Obviously, a large proportion of TM-containing proteins are not at the plasma membrane but embedded in internal organellar membranes. Applying the predictors TMHMM, SignalP 5.0, DeepLoc and IPC (see Table 4 for their characteristics and purposes) to all 8284 *T. gondii* proteins leaves only 29, of which only four are SAG-related surface (SRS) proteins known or assumed to be surface-located [79], plus one bradyzoite cyst wall protein recently identified by BioID labeling [104]. With the simple filter function in Excel one can compile one’s own list of candidates. However, the caveats of this rather naive approach become obvious with another example.

As noted above, a total of 111 proteins in ToxoDB are annotated as SRS proteins due to sequence homology. This would imply that they contain C-terminal sequence motifs consistent with being GPI-anchored. The GPI-predictor, NetGPI 1.1, identifies 95 of them as having this anchor. Almost all known GPI-anchored proteins possess a signal peptide due to the GPI synthesis pathway being located in the ER [126]. However, from 321 NetGPI-positive proteins only 95 are SignalP 5-positive, which raises the question whether only these 95 proteins are real GPI-anchored proteins. When both predictors are applied (SignalP and NetGPI) only 47 of the 111 SRS proteins (42%) are left. One could suspect SignalP 5.0 for missing some signal peptides, and using TargetP (non-plant setting; 838 total signal peptides) instead gives indeed different results (70 SRS, or 63%), whereas DeepSig (548 signal peptides predicted) only recognizes 33% of the GPI-positive SRSs as having a signal peptide. Finally, all predictors together identify 36 proteins (33%) as signal peptide- and GPI anchor sequence-containing SRS (Figure 1A). Only seven SRS proteins “resist” any placement. It should also be noted that most training sets for the different algorithms do not use (m)any proteins from protozoa, which in turn might influence their sensitivity and/or specificity. Taken together, in silico predictions are certainly useful but potentially leave out numerous true candidates or make wrong assignments.

### 3.3. Correlation between Tachyzoite mRNA and Protein Abundance—Indirect Quantitative Measures for Oocyst Proteomes?

Several studies have reported a positive correlation between serological recognition and antigen abundance (reviewed in [96]), including *P. falciparum* merozoite antigens [129]. However, genome-scale protein determinations are non-trivial, in particular for less-accessible parasite stages such as the oocysts. Moreover, quantification of membrane proteins is particularly challenging [130]. Therefore, a widespread alternative approach is quantitative transcriptomics [131,132]. Although it is well known that mRNA abundance does not necessarily reflect protein abundance, it holds true for a majority of cognate proteins and transcripts [131], also exemplified by a recent study of *P. falciparum* [133]. When we analyzed the published ribosome profiling data (as a measure for transcript abundance being translated) from *T. gondii* tachyzoites [128,134] with the quantitative proteomics data from the hyperLOPIT study [105] (Figure 1B) a very good correlation could be observed (*R* = 0.68; *p* < 2.2^−16^), given that the data were from different experimental studies. Protein abundance is also dependent on mRNA stability. Recently, bias of GC content at the 3rd (wobble) position of codons (GC3) has been found to influence mRNA abundance and in turn protein abundance in trypanosomatides [135]. Stability of mRNA in humans is also reported to be connected to GC3 bias [136,137], and in general codon usage affects mRNA stability in a range of organisms [138]. We find a moderately good correlation between GC3 and the ribosome profiling data [128,134] (Pearson’s *R* = 0.37, *p* < 2.2^−16^; Figure 1C). In the absence of robust quantitative protein data for oocysts transcript abundance obtained by RNAseq as well as GC3 or other values for codon usage could both serve as a proxy for protein abundance, an important factor for a candidate antigen.

### 3.4. Genome-Wide Prediction of Linear B-Cell Epitopes

It is generally assumed that B-cell epitopes are mostly conformational (discontinuous) since antibody-epitope interactions comprise multiple contacts with sequential segments or single residues that are in close proximity in the folded structure of the antigen [139]. However, prediction of conformational epitopes requires three-dimensional structures of the protein of interest and is thus not suitable for the genome-wide prediction of such epitopes. This may change in the future due to the recent description of two extremely accurate programs for artificial intelligence-based ab initio 3D-structure prediction, AlphaFold [140] and RoseTTAFold [141].

This leaves linear B-cell epitopes and their in silico prediction as a current option. A recent analysis of 488 such structures concluded that only 4% were truly linear (13.5% when up to six interspersed single residues within the epitope were allowed) [139]. The web-based linear epitope prediction program BepiPred 2.0 [142] is considered to be one of the most-advanced tools presently available. Nevertheless, a recent comparison of current algorithms aimed to predict linear B-cell epitopes, including BepiPred 2.0, concluded that “the performance of all predictors (…) was found marginally better than random classification” [143]. This is consistent with our own experience applying BepiPred 2.0 on the *T. gondii* dataset, which showed little correlation between positive epitope scores and known antigens. Nevertheless, in Appendix A we included three genome-wide predictions of antigenicity indices for comparison purposes.

### 3.5. Where Does This All Lead Us with Regard to the Prediction of Stage-Specific Antigens?

It is apparent that in silico predictions can only be seen as a supplement to experimental data, reaffirmation of candidate antigen choice in addition to other criteria. Another way to apply in silico predictions, or data obtained from tachyzoite studies would be to use them as exclusion criteria for other infective stages, such as bradyzoites and oocysts, which are hard to obtain in large and/or pure enough numbers for experimentation. For instance, transcripts that are consistently found in the numerous RNAseq tachyzoite datasets available have a high probability to be not bradyzoite- and/or oocyst-specific. Also, the genome-wide CRISPR/Cas9 study for almost all *T. gondii* genes [144] resulted in 3376 genes (see Appendix A) that showed negative growth phenotypes in tachyzoite culture (having a fitness score ≤ −1.25) [145], which disqualifies them for stage-specificity. On the other hand, those without a negative fitness score, when combined with other data indicating their absence in tachyzoites, provides further evidence that they could be truly stage-specific. Furthermore, many proteins identified as being present in sub-compartments by either the hyperLOPIT experiments [105] or organellar sub-proteomes [146] can also be used as exclusion criteria when the focus is on extracellular antigens being likely candidates for presentation to the immune system. This approach allows us to compile smaller, more manageable subsets of potential stage-specific antigens that can then be further tested by antigen arrays or even by individual recombinant expression, and subsequent serological verification.

## 4. Verification of Protein Candidates to Identify Oocyst-Driven *T. gondii* Infections: The Rationale under a Proposed Workflow for Future Progress

The few studies conducted so far that tried to differentiate the route of *T. gondii* infection by serology based on specified antigens had some relevant limitations (Table 1, Table 2 and Table 3). First, there exists no reference test (gold standard) and the data on protein antigenicity in the various different host species are scarce. Second, the experimental designs are quite heterogeneous, including diverse host species, procedures and inoculation doses for experimental animal infections, parasite stage/strain used and inoculation doses, time points studied and selection criteria of animal and human sera analyzed. These diverse variables can greatly influence immunological parameters and outcomes. Third, in some cases few details were provided on the recombinant target proteins employed in the various immunological assays, in particular on the expression system used, antigen purity, its structural integrity and its resemblance to the native parasite protein. Two main issues arise when designing a verification scheme for novel candidate antigens: the choice of reference sera from animals and humans employed, and the absence of an appropriate and commonly accepted reference test.

Sera from experimental infections are preferred in order to work with a well-defined panel of reagents since in the absence of a reference test it is easier to show that these animals were truly infected. In contrast, naturally infected animals are not just exposed to oocysts, and thus stage-specific immune responses cannot be taken for granted. Moreover, time p.i. is difficult to estimate, which further complicates the interpretation of the results. Despite these caveats, several authors have used sera from farmed pigs and chickens and assumed an infection due to a previous oocyst exposure [63,64].

The mouse model has been extensively employed to carry out experimental infections with different *T. gondii* strains and parasite stages due to easy handling, short-term experiments and low cost. However, as the immunological background of mice and humans differ, results might not be easily extrapolated and should be corroborated in other species more similar to humans. In this sense pig sera could be ideal since this animal shares with humans 80% of genes involved in immune responses, opposed to only 10% of such genes being conserved in the mouse [16].

In order to be able to compare oocyst-driven infections vs. tissue cyst-driven infections the study needs to demonstrate an oocyst-specific response as for ERP [62]. In addition, monitoring of clinical signs and IgG/IgM kinetics during experimental infections deliver valuable data to select well-defined sera.

Finally, the immunogenicity of candidates should be assayed with human sera, as attempted for ERP and CCp5A. A major obstacle consists in the scarce availability of sera from humans with defined tissue cyst-driven infections that may serve as controls. Moreover, sera considered to derive from infections by oocysts usually came from endemic settings with high environmental contamination but where meat-borne infections could not be ruled out [50,62,64]. A complete epidemiological and clinical history should accompany tested serum samples, together with serological data on the phase of the infection. As outlined in Figure 2, sera from non-infected humans and from cases identified in waterborne outbreaks should ideally be tested. Given these limitations the serological studies that advocated ERP or CCp5A as diagnostic antigens for the detection of oocyst-driven infections in humans should be interpreted with caution.

The absence of reference tests (i.e., a test based on a mixture of native oocyst/sporozoite antigens) hampers the selection of adequate control sera and the estimation of diagnostic performance. Only one study reported the recognition of sporozoite antigens in Western blot by human and pig sera from seropositive individuals [62]. However, the pattern of antigen recognition was not mentioned and the diagnostic value of the Western blot with a well-defined sera panel and whether this method could be used as an additional reference test for sera characterization is unknown. Potential differences in the recognition of native oocyst/sporozoite-derived proteins by Western blot between sera from animals infected with either oocysts or tissue cysts has yet to be explored and remains a difficult task. Accordingly, a combination of available tests is recommended to characterize control sera. Commercial ELISAs have been employed in addition to a Western blot with tachyzoite proteins, as a confirmatory or complementary test [147,148]. Likely candidates should be screened by Western blots, with appropriate control sera, prior to developing a proof-of-concept ELISA. These should prove the absence of antigen recognition by sera from animals infected with tissue cysts [62,64] as well as absence of cross reactivity with closely related apicomplexan parasites. False positive results with a commercial test due to the likely presence of cross-reacting antibodies directed against GRA8 and GRA7 proteins from a previous exposure to other Sarcocystidae parasites have been reported [149]. In sheep, false positive results due to cross-reactions with *N. caninum* SAG1 and the chimeric SAG1-GRA8 as antigens were documented [150]. Thus, the possibility of false positive reactions with oocyst/sporozoite-specific proteins cannot be excluded, in particular with proteins highly conserved within the Sarcocystidae, such as members of the OWP family. Ideally, in a further step the use of a panel of well-described samples should estimate cut-offs and the diagnostic performance of these assays, which should also be validated with human sera.

Finally, an important requirement frequently not observed is testing of the assay by ring trials. According to OIE guidelines for the validation of diagnostic assays [151], after analytical (stage 1), diagnostic (stage 2) and reproducibility characteristics (stage 3) have been determined, the diagnostic test should be deployed to other laboratories (stage 4). The same principles apply to human diagnostic test development [152]. At this stage, the availability of reference standards and a commonly accepted interpretation of results are critical. Reproducible results allow testing the applicability of the method in surveillance and monitoring trials.

Based on these discussed considerations and limitations we propose a workflow to develop an assay to differentiate oocyst-driven infections vs. tissue cyst-driven infections (Figure 2).

## 5. Conclusions

In this article we have described the current knowledge of *T. gondii* proteins that have been reported to be useful serological antigens to identify oocyst-driven infections. As outlined, such diagnostics are essential tools for source attribution of *T. gondii* infections in the One Health context and for devising strategies to limit the impact of environmental oocyst contamination where possible. While the oocyst wall and sporozoite proteins described in the literature for this purpose appear promising, we also highlighted several procedural issues that require further attention. We proposed a schematic workflow that included the in-silico identification of candidate antigens and their validation by strictly defined experimental animal sera before they should be considered as robust diagnostic antigens. While our approach focused on antigenic oocyst-specific proteins, a similar procedure could also be used to identify new bradyzoite-specific antigens.

## Figures and Tables

**Figure 1 microorganisms-09-02346-f001:**
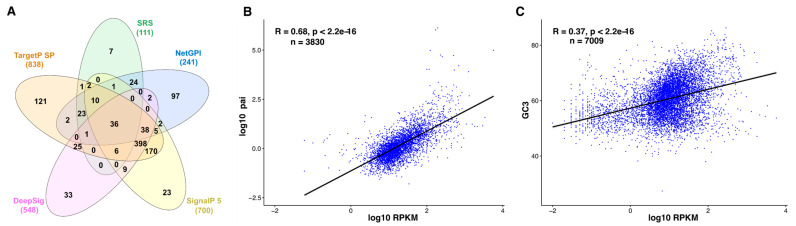
In silico analyses of *Toxoplasma gondii* proteins. (**A**) Venn diagram [127] illustrating the number of SRS proteins with respect to having either a signal peptide and/or a GPI anchor, as predicted by different algorithms. For details see text. (**B**) Genome-wide correlations between protein and RNA abundance in *T. gondii.* Correlation in tachyzoites between mean of transcript levels (expressed as log10 of reads per kilobase of transcript per million mapped reads, RPKM) and protein abundance (expressed as log10 of protein abundance index, PAI) reported by [105,128]. Analysis is based on 3830 genes/proteins shared by both data sets. (**C**) Similar correlation analysis with GC3 calculated from 7009 gene data shared by both data sets, using CodonW 1.4. Statistical analysis was performed with the R package ggpubr 0.4.0., reporting the Pearson correlation coefficient.

**Figure 2 microorganisms-09-02346-f002:**
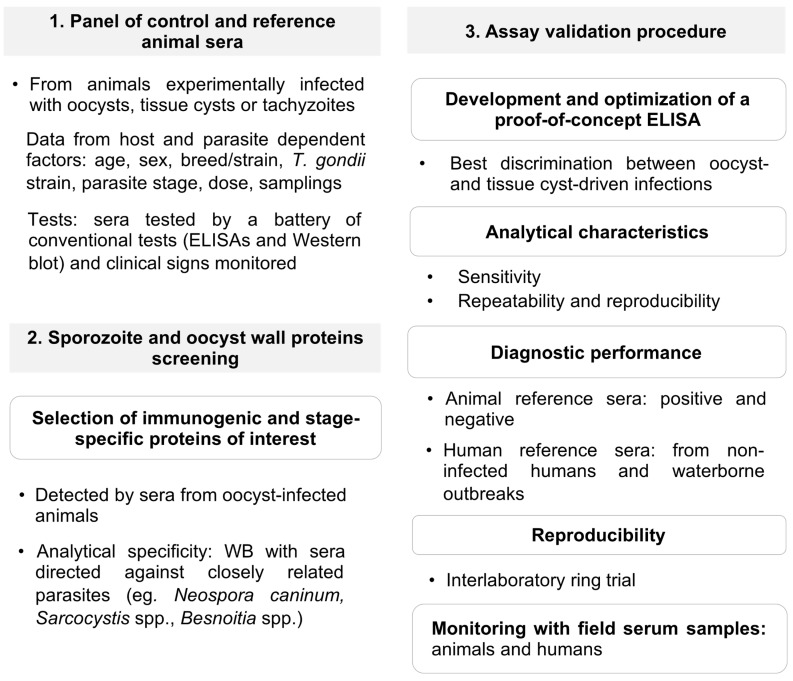
Workflow to develop an assay to differentiate oocyst-driven infections vs. tissue cyst-driven infections from a One Health perspective and according to OIE guidelines.

**Table 1 microorganisms-09-02346-t001:** Oocyst wall recombinant protein-based serological tests developed to differentiate oocyst-driven infections from tissue cyst-driven infections in animals.

Proteins (MW)	Assays Developed *	Ref. Test	N/E	Host-Dependent Factors					Parasite-Dependent Factors			Results	Ref.
				Animal Species	Nº	Age	Sex	Samplings	*T. gondii* Strain	Stage: TZ, TC, Oo	Dose		
**OWP1** (40KDa)	ELISA/WB	Soluble TZ ELISA	N	Pig	44	na	M	1 sampling	-	- *	-	OWP1: no recognition	[64]
			E	Pig	3	6.5–7.5 w	F and M	28 dpi	RH	TZ	10^6^	OWP1: no recognition	
			E	Pig	3	6.5–7.5 w	F and M	28 dpi	VEG	Oo	1.5 × 10^4^	OWP1: no recognition	
			N	Chicken	113	na	na	1 sampling	-	- **	-	OWP1: recognition OWP1 ELISA: 74% ^+^ TZ ELISA: 80% ^+^	
			E	Chicken	na	na	na	1 sampling	RH	TZ	100 µg (two 15d-interval boosters)	OWP1: no recognition	
			E	Chicken	na	na	na	1 sampling	VEG	TC	100	OWP1: no recognition	
**OWP8** (65KDa)	ELISA/WB	recGRA7 ELISA	N	Pig ***	90	na	na	1 sampling	-	-	-	OWP8 ELISA: 12.2% ^+^ GRA7 ELISA: 16.7% ^+^	[63]
			N	Chicken	96	na	na	1 sampling	-	-	-	OWP8 ELISA: 13.5% ^+^ GRA7 ELISA: 10.4% ^+^	

MW: molecular weight; Ref.: reference; N: natural infection; E: experimental infection; Nº: number; F: female; M: male; TZ: tachyzoites; TC: tissue cysts; Oo: Oocysts; WB: Western blot; rec: recombinant; w: weeks; d: days; na: no data available; dpi: days post-infection; +: seropositive; -: unknown. * Assays based on recombinant proteins. ** Authors claimed that these animals could possibly be infected with oocysts; Animal breed or strain were unknown in all studies. *** 15 *T. gondii* tachyzoite-positive porcine sera did not recognize OWP8 by WB.

**Table 2 microorganisms-09-02346-t002:** Oocyst recombinant protein-based serological tests developed to differentiate oocyst-driven infections from tissue cyst-driven infections in humans.

Proteins (MW)	Assays Developed *	Ref. Test	Cohorts Studied							Results	Ref.
			Nº	Age	Sex	Origin	Seropositive (Tests)	Other Relevant Data	Samplings		
**Oocyst Wall Proteins**											
**OWP8** (65 kDa)	ELISA/WB	recGRA7 ELISA	169	na	na	Hospital	na	-	One sampling	OWP8 ELISA: 3.6% ^+^ GRA7 ELISA: 14.7% ^+^	[63]
**Sporozoite Proteins**											
**ERP** (11 kDa)	ELISA/WB	Conv. tests	6	na	na	Laboratory employees	yes (IgM/IgG WBs)	Outbreak	From 1 m post exposure till 8 mpi	ERP: recognition by 100% by WB at 1 mpi and detectable Abs for 5–6 mpi **	[62]
			4	na	na	Chronically infected people (IgM− and IgG+)	yes (dye test and IFAT)	-	One sampling	ERP WB: no recognition	
			11	na	na	Visitors to a horse stable	yes (IgM/IgG^+^, dye test/IFAT)	Outbreak	Between 78 and 149 days after onset of symptoms	ERP WB: 82%^+^	
			182	18–43 yr	na	Settings where the infection is prevalent	yes (IgG ELISA/avidity ELISA)	-	One sampling	ERP WB: 60^+^ (29 and 31 acute and chronic infections) ERP ELISA: 44^+^ (23 and 21 acute and chronic infections)	
			10	adults and two 2 yr siblings	na	8 sera from Amish family (1 serum from a congenitally infected child)	yes (IgA, IgM and IgG conv. tests) with the exception of 1 sibling	-	One sampling	ERP WB: 6 adults ^+^	
			76	na	F	Congenital infections	yes (IgG/IgM^+^)	3 mothers acquired acute toxoplasmosis during an outbreak	2.5 m after childbirth	ERP WB: 78% + 2.5 m after birth	
			59	na	F	Chronically infected people	yes (58 IgG^+^; 1 IgG^+^ and IgM^+^)	-	One sampling	WB ERP: no recognition	
	ELISA	Sabin Feldman Dye test and ELISA	10	na	na	Blood donors	yes (seroconversion in 2nd sampling)	-	Two consecutive samplings	ERP ELISA: 1^+^ no recognition in 2nd sampling	[74]
	ELISA	Conv. IgG/IgM tests	476 *** 380	0–28 yr	177M/299F 161M/219F	Endemic area of toxoplasmosis Public schools (students, parents, school staff)	yes (249 IgM^+^ and/or IgG^+^)	Serum and saliva pairwise comparisons ****	One sampling	Divergent results among conventional ELISA and TgERP ELISA for saliva and sera in all age groups. Prevalence values similar (between 66.6–68.7%) for both ELISAs in 15–21 yr age group	[75]
	ELISA	Conv. IgG/IgM tests	128	na	na	Areas with groundwater vulnerability (unconfined aquifers)	111 (IgG^+^) All individuals: IgM^+^	-	One sampling	ERP ELISA: 63^+^ >OD values in younger people	[50]
**CCp5A** (50 kDa)	ELISA/WB	Conv. tests	78	na	na	Outbreak	yes	Acute clinical signs, ocular disease; attributed to contaminated water	One sampling	CCp5A ELISA: higher IgG/IgM levels than in pregnant women TZ ELISA: higher IgM levels than in pregnant women	[64]
			78	na	F	Pregnant women	yes (IgG^+^ and IgM^+^)	-	One sampling	CCp5A ELISA: lower IgG/IgM levels than in the outbreak; 80% IgM^+^ (evidence of recent exposure to *T. gondii*) TZ ELISA: IgG levels higher than in the outbreak	
**CCp5A** (17 kDa fragment)	ELISA/WB	recGRA7 ELISA	169	na	na	Hospital	na	No previous clinical/serological data	One sampling	CCp5A ELISA: 3% ^+^ GRA7 ELISA: 14.7% ^+^	[63]
**SporoSAG** (25.6 kDa)	ELISA	recSAG1 ELISA/recSRS2 ELISA	13	na	na	Waterborne transmission	yes (SAG1^+^)	-	One sampling	SporoSAG: no recognition by anti IgA, IgM and IgGs	[76]
			1	na	na	Infected with type II strain oocysts (control serum)	yes	-	One sampling	SporoSAG: no recognition by anti IgA, IgM and IgGs	
			6	na	F	Pregnant women	yes	Presumably infected by meat route	One sampling	SporoSAG: no recognition by anti IgA, IgM and IgGs	

MW: molecular weight; Ref.: reference; Nº: number; F: female; M: male; WB: Western blot; IFAT: indirect immunofluorescence antibody test; Conv.: conventional; rec: recombinant; yr: year; na: no data available; -: unknown.* Assays based on recombinant proteins; ** In one employee anti-TgERP antibodies were detected until 8 mpi; No sequential samples were collected with the exception of laboratory workers [61] and blood donors [74]; *** 476 sera; 380 serum-saliva paired samples; **** Difficulty in considering a positive IgG and/or IgM reaction as criteria of TgERP seropositivity and infection since seronegative but infected individuals might give a positive TgERP result.

**Table 3 microorganisms-09-02346-t003:** Sporozoite recombinant protein-based serological tests developed to differentiate oocyst-driven infections from tissue cyst-driven infections in animals.

Proteins (MW)	Assays Developed *	Ref. Test	Host-Dependent Factors						Parasite-Dependent Factors			Results	Ref.
			N/E	Species (Strain)	Nº	Age	Sex	Sampling Period	*T. gondii* Strain	Stage: TZ, BZ, TC, Oo	Dose		
**ERP** (11 kDa)	ELISA/WB	MAT/ELISA	E	Pig	10	5 m	na	9 mpi	VEG	Oo	1000	ERP WB: recognition for 6–8 mpi	[62]
		MAT/ELISA	E	Pig	10	5 m	na	9 mpi	VEG	TC	5000	ERP WB: no recognition	
		-	E	Mice (Swiss Webster)	na	na	na	60 dpi	ME49	Oo	50	ERP WB: recognition	
		-	E	Mice (Swiss Webster)	na	na	na	60 dpi	ME49	TC	na	ERP WB: no recognition	
	ELISA	IgG/IgM tests	N	Chicken	198	na	na	1 sampling	-	- **	-	ERP: recognition by 49%	[50]
**CCp5A** (50 kDa)	ELISA/WB	Soluble TZ ELISA	N	Pig	44	na	M	1 sampling	-	-	-	CCp5A ELISA: 100% ^+^ TZ ELISA: 100% ^+^	[64]
			E	Pig	3	6.5–7.5 w	F and M	28 dpi	RH	TZ	10^6^	CCp5A WB: no recognition TZ ELISA: seroconversion	
			E	Pig	3	6.5–7.5 w	F and M	28 dpi	VEG	Oo	1.5 × 10^4^	CCp5A ELISA: recognition; Abs peaked at 7dpi and decrease at 28dpi TZ ELISA: seroconversion	
			N	Chicken	113	na	na	1 sampling	-	-	-	CCp5A ELISA: 70% ^+^ TZ ELISA: 80% ^+^	
			E	Chicken	na	na	na	1 sampling	RH	TZ	100 ug (two 15d boosters)	CCp5A WB: no recognition	
			E	Chicken	na	na	na	1 sampling	VEG	TC	100	CCp5A WB: no recognition	
			E	Mice (Balb/c)	5	8–12 w	na	60 dpi	VEG	Oo	50	CCp5A WB: recognition anti CCp5A IgMs peaked at 15 dpi TZ ELISA: seroconversion	
			E	Mice (Balb/c)	5	8–12 w	na	60 dpi	VEG	TC	50	WB/ELISA CCp5A: no recognition TZ ELISA: seroconversion	
**CCp5A** (17 kDa fragment)	ELISA/WB	recGRA7 ELISA	N	Pig ***	90	na	na	1 sampling	-	-	-	CCp5A ELISA: 12.2% ^+^ GRA7 ELISA: 16.7% ^+^	[63]
			N	Chicken	96	na	na	1 sampling	-	-	-	CCp5A ELISA: 9.4% ^+^ GRA7 ELISA: 10.4% ^+^	
**SporoSAG** (25.6 kDa)	ELISA	recSAG1 ELISA/recSRS2 ELISA	E	Mice	1	na	na	1 sampling	ME49	Oo	100	SporoSAG ELISA: no recognition by anti IgA, IgM and IgG antibodies.	[76]
			E	Mice	3	na	na	1 sampling	ME49	Oo ****	100	SporoSAG ELISA: no recognition by anti IgA, IgM and IgG antibodies.	
			E	Mice	4	na	na	1 sampling	ME49	BZ	20	SporoSAG: no recognition by anti IgA, IgM and IgG antibodies.	
**SporoSAG** (23.78 kDa)	ELISA	recGRA1 ELISA	E	Mice (Swiss strain)	6	na	na	120 dpi	PRU	Oo	8–10	SporoSAG ELISA: IgM peaked at 1, 10 and 15 dpi IgG peaked at 40 and 120 dpi	[65]

MW: molecular weight; Ref: reference test; N: natural infection; E: experimental infection; Nº: number; F: female; M: male; TZ: tachyzoite; BZ: bradyzoite; TC: tissue cysts; Oo: Oocysts; WB: Western blot; rec: recombinant; m: months; w: weeks; mpi: months post-infection; dpi: days post-infection; na: no data available; ND: not determined; +: seropositive; -: unknown; Abs: antibodies.* Assays based on recombinant proteins; ** Authors claimed that these animals could possibly be infected with oocysts; Animal breed or strain were unknown in all pig and chicken studies; *** 15 *T. gondii* tachyzoite-positive porcine sera did not recognize OWP8 by WB; **** Oral infection (*n* = 1); subcutaneous infection (*n* = 2).

**Table 4 microorganisms-09-02346-t004:** An overview of the characteristics and purpose of different prediction programs used in this study *.

Program	Purpose/Prediction	Link **	Reference
TMHMM 2.0	transmembrane regions	https://services.healthtech.dtu.dk/service.php?TMHMM-2.0	[114]
TOPCONS 2	transmembrane regions	https://topcons.net/	[115]
SignalP 5.0	signal peptide	https://services.healthtech.dtu.dk/service.php?SignalP	[117]
TargetP 2.0	N-terminal sorting signals	http://www.cbs.dtu.dk/services/TargetP/	[118]
NetGPI 1.1	GPI anchor	https://services.healthtech.dtu.dk/service.php?NetGPI	[119]
PredGPI	GPI anchor	http://gpcr2.biocomp.unibo.it/gpipe/index.htm	[120]
GPS-Lipid	lipid modifications	http://lipid.biocuckoo.org/	[121]
DeepLoc	intracellular localization	http://www.cbs.dtu.dk/services/DeepLoc-1.0/	[108]
BUSCA	intracellular localization	http://busca.biocomp.unibo.it/	[107]
Isoelectric Point Calculator (IPC)	consensus pI value	http://isoelectric.org/	[122]
CodonW 1.4	GC3 (GC content 3rd codon position)	http://codonw.sourceforge.net	NA
VaxiJen 2.0	protein immunogenicity prediction	http://www.ddg-pharmfac.net/vaxijen/VaxiJen/VaxiJen.html	[123]
iBCE-EL	linear B cell epitope prediction	http://thegleelab.org/iBCE-EL/iBCE.html	[124]
Secret-AAR	abundance of antigenic regions	http://microbiomics.ibt.unam.mx/tools/aar/	[125]

* For details of programs/methods see respective reference. For calculated values for each *T. gondii* protein see Appendix A. ** Accessed on 4 October 2021.

## Data Availability

All data are included in the text or Appendix A.

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
