# Peer review of "Identification of Oocyst-Driven Toxoplasma gondii Infections in Humans and Animals through Stage-Specific Serology—Current Status and Future Perspectives"

_microorganisms, 2021, doi:10.3390/microorganisms9112346_

Round 1
Reviewer 1 Report
This review paper focuses on a fundamentally important yet understudied field of toxoplasmosis research: discriminating route of infection using serology. The paper is well written, timely and provides a significant contribution in terms of a comprehensive review of the topic to researchers and medical professionals. My primary comment is that the paper is quite long, and the authors should consider areas that can be removed or condensed without losing substance. For example, text re. chronicity of infection does not seem directly relevant to the core objective of the review and distracts from the important content that does focus on stage-acquired diagnostics. Some additional specific comments are provided below.
Specific comments:
- page 2 94-96 Re this statement “Changing the litter box of a cat or petting dogs that have rolled in cat feces are among putative sources of oocyst-driven infections in humans, exemplifying that this infection route does not necessarily involve direct cat contact”
=> These routes are likely less important than oocyst-borne infections from soil/food/water contaminated by oocysts from outdoor cats or pet cats that defecate outside. Cats using litter boxes are less likely to be as important as shown in many studies… consider using more likely scenarios for oocyst borne routes. Or just delete since sentence you mention these other more likely routes later in that section.
2. page 4 – re. the section in the intro on serology diagnostics for acute vs chronic infections: Based on the title and abstract of the paper, this question on timing of infection does not appear to be the main objective of the review – can this section be omitted or written more concisely to focus on how it relates to route of infection serodiagnostics? E.g. questions on avidity testing in animals does not seem directly relevant – if it is please explain why.
Table 1: IgM and IgG in animals – this information does not appear very relevant to focus of paper, consider clarifying relevance, put in SI, or remove.
3. page 16 line 337 might want to mention new peptide array approaches for serotyping
4. pages 17-18 3.2. Re “In silico prediction of the surfaceome – _illustrating the limits”
this section includes great material but quite lengthy – could probably trim to ~50-75% of this text without losing core content
5. Table 5 could be in SI
6. 20 Re. “3.3. Correlation between tachyzoite mRNA and protein abundance - indirect measures for oocyst proteomes?
Not sure this section adds substantial review material that is essential to the core aim of the paper – if there is a relationship between these data and selection of proteins for stage-specific seroassay diagnostics, please clarify.
Figure 2 – same comment as above
Overall this is a great review paper and provides a solid overview for current state of knowledge as well as well-founded suggestions for steps needed to advance reliable seroassays that can discriminate different routes of Toxoplasma transmission.
Author Response
Responses to comments from the Reviewers
Reviewer 1
This review paper focuses on a fundamentally important yet understudied field of toxoplasmosis research: discriminating route of infection using serology. The paper is well written, timely and provides a significant contribution in terms of a comprehensive review of the topic to researchers and medical professionals. My primary comment is that the paper is quite long, and the authors should consider areas that can be removed or condensed without losing substance. For example, text re. chronicity of infection does not seem directly relevant to the core objective of the review and distracts from the important content that does focus on stage-acquired diagnostics. Some additional specific comments are provided below.
Thank you for recognizing our paper as valuable, timely and well written, and for the useful comments and suggestions. We have considered and addressed all the comments and revised the manuscript accordingly.
We have condensed the text where possible without losing substance, taking into account the wide variety of possible readers with different backgrounds and interests, and also taking into account the comments from Reviewer 2 that indicated satisfaction with the amount of information.
Specific comments:
- page 2 94-96 Re this statement “Changing the litter box of a cat or petting dogs that have rolled in cat feces are among putative sources of oocyst-driven infections in humans, exemplifying that this infection route does not necessarily involve direct cat contact”
=> These routes are likely less important than oocyst-borne infections from soil/food/water contaminated by oocysts from outdoor cats or pet cats that defecate outside. Cats using litter boxes are less likely to be as important as shown in many studies… consider using more likely scenarios for oocyst borne routes. Or just delete since sentence you mention these other more likely routes later in that section.
Thank you for pointing this out. We deleted the sentence.
- page 4 – re. the section in the intro on serology diagnostics for acute vs chronic infections: Based on the title and abstract of the paper, this question on timing of infection does not appear to be the main objective of the review – can this section be omitted or written more concisely to focus on how it relates to route of infection serodiagnostics?
E.g. questions on avidity testing in animals does not seem directly relevant – if it is please explain why.
Table 1: IgM and IgG in animals – this information does not appear very relevant to focus of paper, consider clarifying relevance, put in SI, or remove.
While we think that the relevance of this section relates to the fact that these serological tools are needed to characterize the different groups of human sera employed for the screening of putative oocyst or sporozoite antigens in combination with epidemiological data, we agree that Table 1 and the respective part in the section about avidity might seem less relevant for some readers. We therefore moved the table into the supplement (now Table S1), deleted part of the main text and moved other parts into the legend of Table S1.
- page 16 line 337 might want to mention new peptide array approaches for serotyping
Peptide arrays are mentioned in the text, including citations to recent papers from 2019 and 2021. As serotyping is not highly relevant to the core objective of the review, this part was kept concise.
- pages 17-18 3.2. Re “In silico prediction of the surfaceome – _illustrating the limits”
this section includes great material but quite lengthy – could probably trim to ~50-75% of this text without losing core content
This part was shortened substantially. Even after this shortening, this part might seem comprehensive, but at the same time it addresses a number of key aspects that many readers of this review might not be overly familiar with and that have not been covered in other papers in this context. We therefore think that quite comprehensive background and examples will make it easier to understand.
- Table 5 could be in SI
This suggestion was considered very carefully. Our conclusion is that moving this table to be a supplement would make it difficult to follow the text, which several times makes reference to the algorithms described in this table. We therefore decided not to move this table to be a supplement.
- 20 Re. “3.3. Correlation between tachyzoite mRNA and protein abundance - indirect measures for oocyst proteomes?
Not sure this section adds substantial review material that is essential to the core aim of the paper – if there is a relationship between these data and selection of proteins for stage-specific seroassay diagnostics, please clarify.
Figure 2 – same comment as above
These comments were carefully considered, and we still think that this analysis is of importance in the context of this review since it suggests an approach that could bypass the challenges to find abundant oocyst surface-associated proteins, which is one determinant of a candidate antigen. We emphasized this aspect further in the text. We incorporated Fig. 2 into Fig.1 (now 1B and 1C), thereby reducing the size of this paragraph.
Overall this is a great review paper and provides a solid overview for current state of knowledge as well as well-founded suggestions for steps needed to advance reliable seroassays that can discriminate different routes of Toxoplasma transmission.
Thank you for this positive feedback! We hope the workflow will prove useful in assessing potential new candidates for stage-discriminating serology, and we are convinced that this review will be useful to a wide variety of readers.
Reviewer 2 Report
In this review , the authors discuss how to succeed in detecting oocyst- driven infections. They described the current knowledge on T.gondii oocyst-derived proteins that may be useful to set up a test to specifically identify oocyst-driven infection. They also propose a workflow to develop that infection.
This article is well written and well documented and can be accepted in present form.
Author Response
Thank you for recognizing our paper as well written and documented. We hope the comprehensive summary of current knowledge and the workflow will prove useful for future efforts setting up stage-discriminating serology.